# Detection of risk for depression among adolescents in diverse global settings: protocol for the IDEA qualitative study in Brazil, Nepal, Nigeria and the UK

Syed Shabab Wahid [ID],[1] Gloria A. Pedersen [ID],[1] Katherine Ottman [ID],[1] Abigail Burgess,[2] Kamal Gautam [ID],[3] Thais Martini,[4] Anna Viduani [ID],[4] Olufisayo Momodu,[5] Crystal Lam,[6] Helen L. Fisher [ID],[2,7] Christian Kieling [ID],[4,8] Abiodun O Adewuya,[9] Valeria Mondelli [ID],[10] Brandon A Kohrt [ID][1]

For numbered affiliations see end of article.

**Correspondence to**
Dr Helen L. Fisher;
helen.2.fisher@kcl.ac.uk

## ABSTRACT

**Introduction** Globally, depression is a leading cause of disability among adolescents, and suicide rates are increasing among youth. Treatment alone is insufficient to address the issue. Early identification and prevention efforts are necessary to reduce morbidity and mortality. The Identifying Depression Early in Adolescence (IDEA) consortium is developing risk detection strategies that incorporate biological, psychological and social factors that can be evaluated in diverse global populations. In addition to epidemiological and neuroscience research, the IDEA consortium is conducting a qualitative study to explore three domains of inquiry: (1) cultural heterogeneity of biopsychosocial risk factors and lived experience of adolescent depression in low-income and middle-income countries (LMIC); (2) the feasibility, acceptability and ethics of a risk calculator tool for adolescent depression that can be used in LMIC and high-income countries and (3) capacity for biological research into biomarkers for depression risk among adolescents in LMIC. This is a multisite qualitative study being conducted in Brazil, Nepal, Nigeria and the UK.

**Methods and analysis** A systematic set of qualitative methods will be used in this study. The Delphi method, Theory of Change (ToC) workshops, key-informant interviews and focus group discussions will be used to elicit perspectives on the study topics from a broad range of stakeholders (adolescents, parents, policy-makers, teachers, health service providers, social workers and experts). Delphi panellists will participate in three survey rounds to generate consensus through facilitated feedback. Stakeholders will create ToC models via facilitated workshops in the LMIC sites. The framework approach will be used to analyse data from the study.

**Ethics and dissemination** Ethical approvals were received from the Ethics Review Board of George Washington University and from site-specific institutions in Brazil, Nepal, Nigeria and the UK. The findings generated from this study will be reported in highly accessed, peer-reviewed, scientific and health policy journals.

### Strengths and limitations of this study

► This study uses the Delphi method, the theory of change approach, key-informant interviews and focus group discussion methods in multiple settings varied by culture, economic status and risk factors to investigate adolescent depression.

► The multisite approach will generate new knowledge on context-specific and universal pathways of adolescent depression manifestation, identification and prevention.

► Perspectives will be elicited from a broad range of stakeholders including adolescents and parents, teachers, health service providers, social workers and policy-makers.

► Due to current ethical restrictions surrounding research with children, independent adolescent perspectives will not be available from Nigeria. Adolescent interviews will not be conducted in the UK either.

► In qualitative focus groups and interviews, individuals with the highest concern may be the most vocal, and thus, the results may over-represent concerns regarding the risk calculator tool.

## INTRODUCTION

Globally, depression is a leading cause of illness and disability among adolescents aged 10–19 years old.[1] The presence of depression also increases the risk of suicide, which is the third-leading cause of death in this population.[1] As the incidence of depression peaks in adolescence, and often remains undiagnosed, the negative consequences of depression persist as a chronic condition throughout the life course.[2] Major depressive disorder, the most commonly diagnosed form of depression, has a lifetime prevalence of 11% representing a major cause of disability across the world.[3]

On a global scale, treatment alone is insufficient to address this problem. Limited efficacy of available interventions along with the limited availability and low-quality of mental health services in many parts of the world persist as significant barriers.[1 4 5] Therefore, increased identification of depression early in adolescence and administering preventive strategies become highly salient in addressing this global burden.[5] Understanding the biopsychosocial risk factors that can predict the onset of depression, and the protective factors which can inform measures for preventing its manifestation and severity, are important steps towards achieving this goal.

Research on depression among adolescents in Western Educated Industrialised Rich and Democratic countries is insufficient to understand the full scale of the problem.[6] Identification and prevention measures need to account for the heterogeneity of risk and protective factors and signs and symptoms of depression, as they manifest across cultures and countries.[7] Therefore, investigating depression across multiple global settings can provide clearer insight into the universal and context-specific risk and protective factors.

Currently, there are three major gaps in global research to address prevention and early identification among adolescents. First, there is a need for context-specific understandings of mental health, adolescent experiences and risk and protective factors. Cultural conceptions of depression or adolescence vary in their meaning and social implications across populations and context. Accordingly, discerning such factors are crucial in informing culturally sensitive depression identification and prevention efforts.

Second, despite a wide range of research on depression, there is a lack of feasible and acceptable tools to determine the risk of depression among adolescents before the disorder develops. Risk scores are important in other fields of medicine to determine when and how to intervene and how best to allocate resources, for example, risk determination and treatment provision for cardiovascular disease, diabetes and cancer.[8–10] Adopting similar tenets to develop a risk calculator for adolescent depression could have substantial public health benefit.

Third, the ethical and institutional policies for research involving adolescents vary widely in low-income and middle-income countries (LMICs). Specifically, ethical policies are often lacking for biological psychiatry research, which is needed to shed light on mechanisms and risk markers for adolescent depression.[11] Biological markers may be an important contributor to risk calculation for depression and to evaluate efficacy of prevention efforts.[12] Therefore, assessing the feasibility of implementing biological psychiatry research and determining its ethical and cultural acceptability is critical to identify necessary infrastructural and policy recommendations to increase LMICs capacity for conducting such studies.

To address these three gaps, this article presents the protocol of a global, multisite, qualitative study which will inform the development of successful adolescent depression identification and prevention initiatives. First,

in our qualitative study, we will investigate the cultural and contextual perceptions and considerations of identifying adolescent depression, including risk and protective factors, and depression prevention interventions. Second, we will explore the feasibility, acceptability and utility of a risk calculator for depression in adolescence. Third, we will investigate the feasibility of conducting biological psychiatry research in LMIC settings. To achieve this, a diverse set of qualitative methods will be used in four countries: Brazil, Nepal, Nigeria and the UK, representing high-, middle-income and low-income settings.

## OBJECTIVES
### Identifying Depression Early in Adolescence research consortium
This qualitative study is part of a larger integrated research portfolio being implemented by the Identifying Depression Early in Adolescence (IDEA) research consortium (https://www.mqmentalhealth.org/research/profiles/identifying-depression-early-in-adolescence).[5] The consortium comprises psychiatrists, epidemiologists, neuroscientists and anthropologists from Brazil, Nepal, Nigeria, the UK and USA conducting multidisciplinary research on adolescent depression identification and prevention. The consortium was established with the support of the MQ charity Brighter Futures initiative. The operational dates of the IDEA study are from November 2018 to November 2021.

### General objective of multicomponent qualitative study
The objective of the current study is to qualitatively explore key topics that will complement epidemiological and biological psychiatry research being conducted through IDEA. For this multicomponent qualitative study, we will consult with adolescents, parents, healthcare providers, social workers, educators, policy-makers and other key stakeholders in Brazil, Nepal, Nigeria and the UK, as well as global experts, about the experience of adolescent depression cross-culturally, the use of a risk calculator for depression during adolescence, and conducting biological psychiatry research with adolescents in LMICs.

### Specific objectives
1. The qualitative component of IDEA will compare cultural and contextual differences in depression, adolescent experiences, health systems and health policy across Brazil, Nepal, Nigeria and the UK. Additionally, information will be collected from an expert panel on risk factors and detection approaches that are specific for adolescent depression and feasible for implementation in LMICs.
2. A risk assessment tool for depression in adolescence has been developed via predictive modelling using cohort data from Brazil, Nigeria, Nepal and the UK.[5 13 14] The tool will determine future risk of depression and is derived from research that classifies risk based on

the presence of sociodemographic risk factors.[13] This qualitative study will explore the feasibility, acceptability, utility and the ethical implications of such a tool.

3. The IDEA research includes biological psychiatry studies with functional neuroimaging and inflammation pathways. The qualitative work will also explore feasibility, acceptability and ethics related to expanding biological psychiatry research in LMIC and incorporating biological markers to evaluate risk of the onset of depression during adolescence.

## STUDY SETTINGS: COUNTRY DESCRIPTIONS

The four IDEA country sites were selected to provide a wide range of contexts to study adolescent depression. Brazil is an upper-middle-income country representing the growing number of countries with rapidly developing economies and urbanisation, collectively known as the Brazil, Russia, India, China and South Africa (BRICS) nations. Nepal is one of the poorest countries in the world and represents conditions of adolescents living in the least-developed nations. Nepal is also representative of the large number of adolescents living in humanitarian settings due to its recent emergence from a protracted civil war, and frequent environmental disasters. Nigeria is a lower-middle-income country and the most populous country in Africa. The experiences of adolescents in Nigeria reflect the rapid development of African economies as well as chronic exposure to political violence, community violence and high rates of infectious diseases, including HIV/AIDS. The UK is the study setting representative of adolescents living in highly resourced regions such as Western Europe and North America. Further details of the countries and country-specific IDEA research teams are provided in online supplementary appendix 1.

## METHODS
### Conceptual framework

The IDEA qualitative study is structured according to the social ecological model of health and Singer and Baer's world system theory on the social origins of disease.[15 16] Using these two guiding theoretical frameworks, we seek to understand the role of individual, interpersonal, institutional, community and policy factors and their interrelations, in depression risk and identification in adolescence (please refer to figure 1). Informed by George Engels' classic model, we will elicit biopsychosocial risk and protective factors of depression within and between each ecological stratum.[17] We will use Kleinman's Explanatory Model framework to explore the lived experience of depression at the individual level, including culturally driven local idioms of distress.[18 19] We will further explore how these explanatory models are influenced by relationships at the interpersonal and primary group levels (family and friends), and cultural and social norms at the community level. At the institutional level, we will examine mental health services for depression identification and management, and acceptability and feasibility of risk detection at schools, primary healthcare and social services. At the policy level, we will seek to understand challenges and opportunities to facilitate better depression detection and management. Additionally, in the LMIC sites, we will explore institutional capacity for conducting biological psychiatry research (ie, biological specimen collection, storage and testing capacity; and research capacity of universities and staff) and policy-level considerations for ethical research governance that can support sensitive biological psychiatry research.

To conduct this study, three methodologies will be used: a Delphi activity, Theory of Change (TOC) workshops and qualitative interviews including both key informant

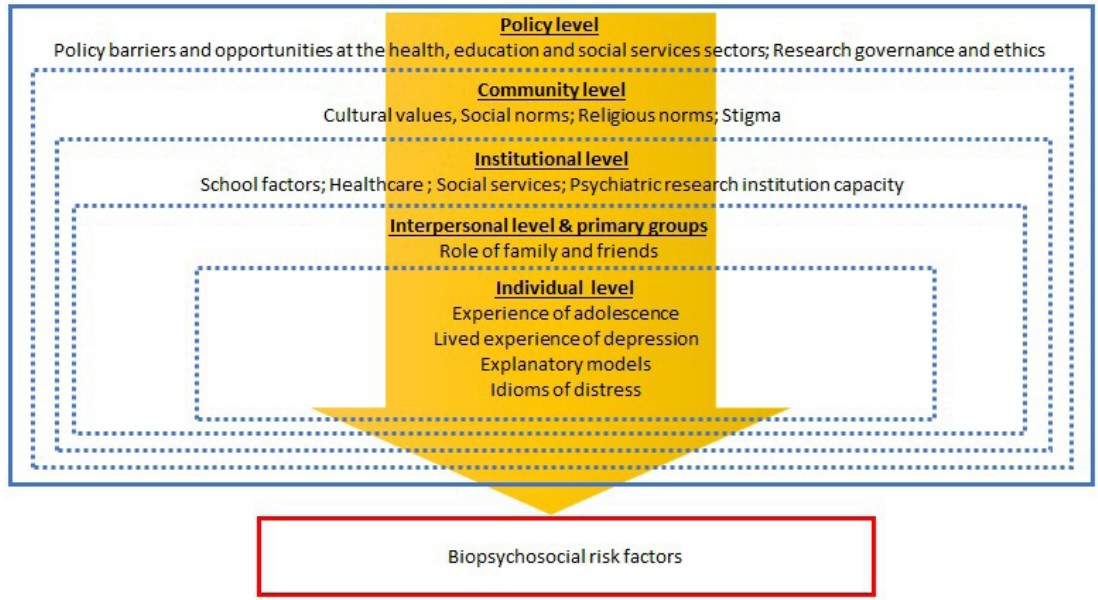

**Figure 1** Conceptual framework for the qualitative component of the IDEA study.

**Table 1** Objectives and methods

| Objectives | Methods | | |
|---|---|---|---|
| | **Delphi activity with international experts** | **Theory of Change workshops in Brazil, Nepal and Nigeria** | **Qualitative interviews in Brazil, Nepal, Nigeria and the UK** |
| Objective 1: Cultural concepts of depression, depression risk and health systems | Experts will comment on risk factors in different cultural settings. | Policy-makers, service providers, parents, adolescents and other stakeholders will develop a theory of change for depression identification and prevention. | Policy-makers, service providers, parents, adolescents and other stakeholders will describe cultural explanatory models of adolescent depression, including help-seeking experiences. |
| Objective 2: Feasibility, acceptability and perceived utility of a risk calculator for adolescent depression. | Experts will comment on feasibility of risk assessment and early detection methods. | Policy-makers, service providers, parents, adolescents and other stakeholders will comment on how to determine depression risk in the theory of change model. | Policy-makers, service providers, parents, adolescents and other stakeholders will review and comment on a mock-up of a risk calculator. |
| Objective 3: Feasibility and acceptability of biological psychiatry research in low-income and middle-income countries. | Experts will comment on specificity and feasibility of risk factors in different cultural contexts. | Policy-makers, service providers, parents, adolescents and other stakeholders will comment on the role of biological research within the theory of change model. | Policy-makers, service providers, parents, adolescents and other stakeholders will describe priorities, barriers and utility of biological research in their settings. |

interviews (KII) and focus group discussions (FGD) (see table 1).

### Delphi activity

A Delphi panel consensus study using quantitative and qualitative methods will be conducted soliciting opinions from global experts in the field of adolescent depression. The Delphi method allows for the systematic generation and scoring of research questions using predetermined criteria, and has been widely used for consensus studies in depression research.[20–25] The Delphi panel will provide a state of the field recommendation on the range and relevance of biopsychosocial risk factors for adolescent depression, strategies for early identification of depression in adolescence and the feasibility for research and preventive interventions, in heterogeneous global settings. Recently, qualitative interviews have been incorporated in Delphi studies to provide meaning to the quantitative results.[26 27]

### Participants

A small international steering committee will be formed to guide the Delphi activity. To identify panellists, key publications in the field will be reviewed and authors will be invited to participate via email. Additional panellists will be identified and recommended by the Delphi steering committee. Respondent profiles will include researchers and academics, clinicians and service providers, and policy-makers from diverse global and economic settings. The sample required for reliability in consensus methodology is between 6 and 15 participants, and this study will aim to establish a panel of 20 global experts.[28]

### Data collection and analysis

The results of a systematic literature review of common biopsychosocial risk factors, a separate component of the IDEA research portfolio, will inform the development of the Delphi questionnaires.[29] Three rounds of surveys will be administered in the Delphi activity. Respondents will generate a list of biopsychosocial risk factors, possible early signs and early detection strategies for adolescent depression in round 1 via an open-ended free listing exercise. The results will be collated via categorisation of similar items into a final list. The finalised items from round 1 will be ranked by panellists according to feasibility and specificity in round 2. From the ranking exercise, measures of centrality will be generated, including frequency, average rank and Smith's salience index, for each item.[30] In round 3, panellists will be provided with these group summary statistics, and given the opportunity to compare and change their rankings, to derive expert consensus. The web-based survey software, Qualtrics, will be used to implement rounds 1–3.[31] Following the three Delphi rounds, we will conduct in-depth interviews with panellists based on quantitative results. Panellists will provide comments on summary results tables explaining personal and panel rankings, interesting patterns in the quantitative results, and provide narratives on how cultural and contextual factors influence biopsychosocial risk factors. Qualitative data will be analysed using thematic analysis.[32]

### Theory of Change workshops

ToC methodology has emerged as a viable alternative to conceptualising programme design and evaluation in global mental health.[33] The ToC method offers a theory-driven approach which identifies and lays out causal pathways that lead to the outcome of a programme or process. A ToC explicitly includes the short-term, medium-term and long-term outcomes that lead up to an expected impact, the interventions and indicators that constitute

the programme and measure its progress, and relevant programmatic and contextual assumptions.[34]

## Participants

The ToC workshops will be composed of approximately 6–12 participants and vary according to context-specific needs. In Brazil, we will conduct four ToC workshops with researchers, adolescents, parents and policy-makers. We will conduct three ToC workshops in Nepal: one with adolescents and youth researchers; one with parents/guardians, schoolteachers and service providers (clinical psychologists); and one with clinicians and policy-makers. In Nigeria, we will conduct two ToC workshops with adolescents, parents, non-governmental organisation (NGO) leaders, health service providers, teachers and policy-makers. ToC workshops will not be conducted in the UK because a lot of research on the neurobiological underpinnings of adolescent mental health is already occurring in this context. Rather, the UK team will conduct interviews with global health academics to gain better understandings regarding their views on the barriers and facilitators to conducting such research in LMIC contexts.

## Theory of Change development

For the IDEA project, a ToC approach is well suited to address each of the objectives as it allows country-specific stakeholder groups to conceptualise and create causal pathways of the feasibility and acceptability of conducting biological and psychosocial research and integrating risk and preventative interventions into existing health, education and social services systems of the study sites. Stakeholder driven discussion during ToC workshops facilitated by IDEA researchers will produce visual maps of these pathways in each research site. Soliciting perspectives from diverse stakeholder groups from the socioecological hierarchies of these countries will enable the formulation of integrative, country-specific and global theories of change of these causal processes.

## Key-informant interviews and focus group discussions

In order to achieve the three objectives, we will conduct qualitative KIIs and FGDs at each site. Key informants are individuals with 'great knowledge…who can shed light on the inquiry issues' and in-depth semistructured interviews with key informants will drive the major focus of data collection for this study.[35] FGDs are highly useful in gleaning both commonly shared and divergent views of a group.[36] FGDs are a useful method for collecting insights from relatively homogeneous groups or among those who have a shared common experience, such as the lived experience of depression.[35]

The thematic areas that will be explored in the interviews include the pathogenesis and lived experience of depression and the contextual considerations of depression identification mechanisms. The experience of depression is heterogeneous across the world.[37] Symptoms of anxiety disorder often co-present or can play a

### Box 1   IDEA study risk calculator

The Identifying Depression Early in Adolescence (IDEA) study risk calculator has been developed as a prognostic model using data from the Pelotas 1993 Birth Cohort, a prospective study conducted in south Brazil.[13] The study included every child born in the city that year: all but 16 mothers agreed to take part, resulting in a total cohort of 5249 individuals. Follow-up visits were made at multiple time points; the data for the composite risk score development were collected at ages 15 and 18 assessments, which had retention rates 85.7% and 81.3%, respectively. Using only sociodemographic variables easily obtainable directly from the adolescent at age 15, we developed a risk calculator with good discriminative performance (c-statistic of 0.78) to identify those at high risk for developing major depression at age 18. A total of 11 variables comprised the final model of the composite score: biological sex, ethnicity, drug use, school failure, social isolation, fight involvement, relationship with mother, relationship with father, relationship between parents, childhood maltreatment, running away from home. As part of IDEA project, the calculator has demonstrated good predictive capacity when externally assessed using data from the Environmental Risk Longitudinal Twin Study from the UK, the Dunedin Multidisciplinary and Development Study from New Zealand and former child soldiers and war-affected adolescents in Nepal.[13 14] The calculator is being tested among adolescents in Lagos State, in Nigeria as well.

role in the onset of depression.[38] Accordingly, in the interviews, we will explore a range of negative affective symptoms by adopting an open-ended approach to probing the signs and symptoms of adolescents' experience of depression. We will also explore the biopsychosocial risk and protective factors and structural and social mediators and moderators of depression manifestation and prevention. Specifically, we will gather stakeholder perspectives on the cultural acceptability and feasibility of the implementation of a risk calculator for adolescent depression (please see box 1).

We will also explore the pathways of how stakeholders seek help for depression, and the availability, accessibility and acceptability of treatment modalities. Finally, we will elicit perspectives exploring the pathways that lead towards or prevent successful recovery from depression.

## Participants

We seek to generate a sample consisting of diverse stakeholders that can provide unique perspectives and help constitute an integrated understanding of the phenomena and processes under investigation. We will engage primary stakeholders such as adolescents (those with both a history of depression and those without) and their parents. Secondary stakeholders will be recruited from the socioecological systems and institutions that develop and nurture the environments which shape the experiences of the primary stakeholders. These include schoolteachers and counsellors, health service providers, social workers and policy-makers. Accordingly, we use a purposeful sampling strategy to solicit key informants with direct experience and deep insight of the inquiry topics (eg, adolescents and parents), or due to the strategic

**Table 2** Key informant interviews (KII) and focus group discussion (FGD) stakeholders in the IDEA project

| Stakeholders | Brazil | | Nepal | | Nigeria | | UK* |
|---|---|---|---|---|---|---|---|
| | KII | FGD | KII | FGD | KII | FGD | KII |
| Adolescents | 6 | 1 | 12 | | | 1 | |
| Health service providers | 12 | | 12 | | 12 | | 12 |
| Policy-makers | 6 | | 6 | | 6 | 1 | 12 |
| Parents | 6 | 1 | 6 | 2 | | 1 | 12 |
| Researchers and academics | | | | | | | 12 |
| School teachers and counsellors | 12 | | 12 | | 12 | | 12 |
| Social workers | 12 | | 12 | | 12 | | 12 |

*No FGDs to be conducted in the UK.

positions they occupy in the health and education systems and social services (eg, social workers or policy-makers).[35]

This multiperspective approach will allow triangulation of the data through comparison of insights between, and across, stakeholder groups. For KIIs, a sample of 12 qualitative interviews is sufficient for identification of themes in relatively homogeneous groups.[39] Only those policy-makers who hold relevant roles and expertise will be recruited for KIIs. Each FGD will be composed of approximately 8–12 participants. In each site, the sample will vary according to context-specific priorities. Table 2 presents the various stakeholder groups targeted for KIIs and FGDs.

For each stakeholder group indicated in table 2, relevant inclusion and exclusion criteria apply. Health service providers need to have training and experience working with adolescents and/or mental health. These can include psychiatrists, psychologists, paediatricians and general physicians. The inclusion of mental health specialists in the IDEA study is foundational, as these stakeholders are pivotal in providing care to depressed adolescents. However, as adolescents often present to primary care with depression, specifically with somatic symptomology, the inclusion of paediatricians and primary healthcare physicians are essential, especially in settings with diminished numbers of, and limited access to, mental health specialists.[40] We will include those social workers who have experience in working with adolescents. For educators, we will only include teachers and school counsellors who work with adolescent students and exclude those working with younger children in the school system. Researchers and academics will be recruited based on expertise in adolescent depression. Policy-makers will be included based on relevant linkages to the health, education and social care systems, and the ability to comment meaningfully on the programmatic and policy environments of the country. Finally, to get rich narratives on context-specific lived experience of depression we will recruit adolescents with current or past history of depression. We will also interview adolescents with no history of depression to broaden our understanding of the knowledge and attitudes surrounding depression. Parents of

these two subsets of adolescents will be recruited to gain perspectives of parental attitudes towards, and knowledge of depression, and understand its impact on families.

### Data collection

The KII and FGD guides will be developed using an iterative process. We will first create an interview guide drawing from our conceptual framework and conduct approximately six preliminary KIIs with different stakeholders, across each site. The data from these initial KIIs will be used to revise the guides for contextual sensitivity. The subsequent KIIs and FGDs will be conducted with these revised guides, with the provision for further edits if subsequent findings indicate it to be necessary. Please refer to figure 2 for an overview of the data collection and data analysis process.

All data collection procedures and decisions will be collated into memos to preserve an auditable record of methodological decisions. KIIs and FGDs will be audio recorded with consent (and assent where applicable), and audio recordings will be professionally translated to English (except brazil, where analysis will be conducted in Portuguese) for data analysis. Local terminology for depression and idioms of distress will be included in the local language along with English translations.

Researchers will complete debriefing forms to capture salient ideas, important exchanges, salient events and other features of the interaction during KIIs and FGDs. Debriefing forms are an essential component of the data analysis process and are intended to provide meaning and understanding of the culture, social situation or phenomenon under investigation, and contextualise the interview transcript.[35 41]

### Data analysis

The major deductive themes of the qualitative study are as follows:
► Understanding of developmental, social and health changes in adolescents.
► Understanding of depression in adolescents: symptoms, impact and help-seeking (eg, how can

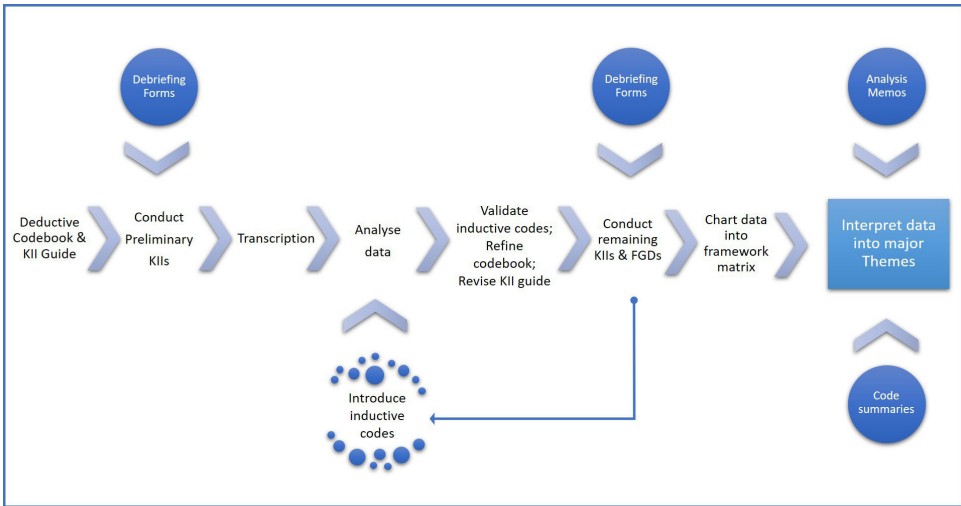

**Figure 2** Data collection and analysis approach. FGDs, focus group discussions; KII, key informant interviews.

depression be detected early and where, whom and when would be best to identify it early).

► Perception of causes or contributory factors to depression in adolescents.

► Views regarding risk detection and possible preventive measures (primary and secondary prevention), including their feasibility, acceptability and utility.

We will use framework analysis (FA) to guide the data collection and analysis of this study[42 43] following the stages of the FA approach in applied qualitative research:
1. Transcription.
2. Familiarisation with the interview.
3. Coding
4. Developing a working analytical framework.
5. Applying the analytical framework.
6. Charting data into the framework matrix.
7. Interpreting the data.

We will adopt a modified approach to implementing the framework approach, as presented in figure 1. The first step will be to create the KII and FGD interview guides and develop a universal codebook that can be used across the four countries. This deductive codebook will be derived from the study objectives, existing literature, theory and expert knowledge. Each country will adjust the codebook to include country-specific codes as necessary. The methodological approach to data collection and analysis will be cyclical and iterative, with each step informing and enhancing subsequent steps, as presented in figure 1.

IDEA researchers across the four country sites will code the data. Codes and themes will be refined using the constant comparison method.[44] This involves moving back and forth between newly coded data and comparing it with previously coded segments to check that the code is still capturing the same essence in the excerpts, as data analysis continues to mature. The addition of new codes and themes will continue until no new codes, categories, or themes emerge from the data, indicating theoretical saturation.[45]

Throughout analysis, researchers will maintain memos to capture ideas, themes, problems, that are extracted from the coding process. Inductive themes can be identified at any stage during the analysis. We will engage in a reflexive approach to data analysis, critically reflecting on the theoretical structures that are drawn out of the analysis.[46] Wherever applicable we will reconceptualise the evidence using other possible theoretical and conceptual models, to test, validate or refine ideas and findings.

Once coding is completed, a summary of each code will be written capturing the essence of that code across the whole dataset, with supporting quotes and researcher insights. We will stratify results according to site and stakeholder characteristics to gain nuanced understandings of how themes vary and converge across groups. In a final step, thematic narratives will be developed in preparation for article manuscripts. For each theme, we will present results from each site to facilitate comparison across countries. We will present similarities and differences in results, across stakeholders and across sites. If applicable, we will construct and present explanatory narratives for any heterogeneity in the results, that is, possibly attributing to cultural or contextual factors. We will use NVivo V.12 software for coding and analysis.[47]

## PATIENT AND PUBLIC INVOLVEMENT

Youth advisory boards have been engaged or established in each of the IDEA countries. The overall IDEA study has been discussed with members of the youth advisory board, and they will be regularly updated on the study progress throughout. The dissemination plan will be developed in accord with the youth advisory boards. Youth advisory boards are not responsible for participant recruitment.

## ETHICS AND DISSEMINATION

The IDEA qualitative study has been reviewed and approved by the institutional review board of the George

Washington University, USA. Additionally, the study has country-specific approvals from the Nepal Health Research Council in Nepal; the Ethics Committee at Hospital de Clínicas de Porto Alegre, in Brazil; the Lagos State University Teaching Hospital Research and Ethics Committee and The Research and Ethics Committee of The Federal Neuropsychiatry Hospital Yaba, Lagos, in Nigeria; and the Psychiatry, Nursing and Midwifery Research Ethics Subcommittee at King's College London, in the UK.

All data collection will operate under the ethical principles of informed consent and assent. Data from the project will be deidentified and stored in password-protected computer servers with access restricted to essential study personnel only. All results will be presented in the aggregate to minimise any potential risks to confidentiality of research participants. Study findings will be reported via publications in academic journals and conferences. In addition to coordination with youth advisory boards, results will also be disseminated through traditional and social media, wherever applicable. We will follow the Consolidated criteria for Reporting Qualitative research guidelines in resultant publications.[48]

## DISCUSSION

In this protocol, we discuss a systematic and progressive set of methods to understand the complex problem of contextual variations in IDEA globally. Through this research we seek to establish the acceptability, feasibility and utility of integrating such approaches in Brazil, Nepal, Nigeria and the UK. Approaches determined to be culturally acceptable, feasible and effective could then be proposed for health providers to support the early detection of depression or identification of risk factors. IDEA and understanding the risk and protective factors of depression would inform policy decisions for funding allocation, amendments to legislation and programming for interventions. Although the prominence of mental health services in health systems varies across countries of different income classifications, compiling stakeholder perspectives can help tailor national mental health policies towards the needs of the specific country. For instance, revelations regarding the effectiveness of existing policies or the level of public awareness about mental illness provide national and community leadership with the context necessary to actualise improvements in current mental health infrastructure.

Investigating these issues using qualitative methods in multiple global settings has not been conducted to date. Therefore, this study will contribute to addressing this gap in the literature. Additionally, there is growing movement and discussion on the transparency, reliability, validity and reproducibility of qualitative research.[49] A systematic presentation of the conceptualisation and procedural implementation of such studies can contribute to the discussion surrounding the transparency and veracity of qualitative methods, and its utility for global mental health research.

The variety of stakeholders to be interviewed presents the opportunity for a range of solutions to be developed across sectors, potentially in the form of public campaigns, health governance restructuring or additions to school resources. Engaging community members and leaders in discussions surrounding adolescent depression is crucial for the sustainability of these changes. By investigating these issues across a range of low-income, middle-income and high-income settings, we hope to generate context-specific, and potentially global understandings of, and responses to, depression among the world's 1.2 billion adolescents.

## Study status

The IDEA qualitative study has finished primary data collection and is in the process of analysing data and preparing manuscripts for publication.

**Author affiliations**
[1]Division of Global Mental Health, George Washington University, Washington, District of Columbia, USA
[2]SGDP Centre, King's College London, Institute of Psychiatry, Psychology & Neuroscience, London, United Kingdom
[3]Transcultural Psychosocial Organization Nepal, Kathmandu, Nepal
[4]Department of Psychiatry, Universidade Federal do Rio Grande do Sul, Porto Alegre, RS, Brazil
[5]Department of Psychiatry, Lagos Island General Hospital, Lagos, Nigeria
[6]Department of Global Health, George Washington University, Washington, District of Columbia, USA
[7]ESRC Centre for Society and Mental Health, King's College London, London, United Kingdom
[8]Child & Adolescent Psychiatry Division, Hospital de Clínicas de Porto Alegre, Porto Alegre, RS, Brazil
[9]Department of Behavioural Medicine, Lagos State University College of Medicine, Lagos, Nigeria
[10]Department of Psychological Medicine, King's College London, Institute of Psychiatry, Psychology & Neuroscience, London, United Kingdom

**Acknowledgements** We thank the members of the youth advisory boards in the four IDEA country study sites.

**Contributors** BAK conceived the paper and was in charge of overall direction and planning. SSW, GAP, KO, AB, KG, TM, AV, OM and CL wrote the manuscript with input from all the authors. CK, HLF, AOA, VM and BAK provided overall technical review, critical revision and final approval for publication.

**Funding** MQ Transforming Mental Health Charity, Brighter Futures grant named 'Identifying Depression Early in Adolescence' [MQBF/1 IDEA]. Additional support was provided by the UK Medical Research Council [MC_PC_MR/R019460/1] and the Academy of Medical Sciences [GCRFNG\100281] under the Global Challenges Research Fund. HLF is supported by a British Academy Mid-Career Fellowship [MD\170005] and the Economic and Social Research Council (ESRC) Centre for Society and Mental Health at King's College London [ES/S012567/1]. CK has received support from Brazilian governmental research funding agencies (Conselho Nacional de Desenvolvimento Científico e Tecnológico [CNPq], Coordenação de Aperfeiçoamento de Pessoal de Nível Superior (CAPES), and Fundação de Amparo à Pesquisa do Estado do Rio Grande do Sul (Fapergs), and is an Academy of Medical Sciences Newton Advanced Fellow. VM has been supported by the National Institute for Health Research (NIHR) Mental Health Biomedical Research Centre at South London and Maudsley NHS Foundation Trust and King's College London. BAK is supported by the US National Institute of Mental Health [K01MH104310, R21MH111280].

**Disclaimer** The views expressed are those of the authors and not necessarily those of the National Health Service, the NIHR, the Department of Health and Social Care, the ESRC or King's College London.

**Competing interests** None declared.

**Patient and public involvement** Patients and/or the public were involved in the design, or conduct, or reporting, or dissemination plans of this research. Refer to the Methods section for further details.

**Patient consent for publication** Not required.

**Provenance and peer review** Not commissioned; externally peer reviewed.

**Author note** The operational dates of the IDEA study are from November 2018 to November 2021.

**ORCID iDs**
Syed Shabab Wahid http://orcid.org/0000-0003-0355-0537
Gloria A. Pedersen http://orcid.org/0000-0003-3427-3464
Katherine Ottman http://orcid.org/0000-0002-8233-8472
Kamal Gautam http://orcid.org/0000-0001-9401-9359
Anna Viduani http://orcid.org/0000-0002-6289-6397
Helen L. Fisher http://orcid.org/0000-0003-4174-2126
Christian Kieling http://orcid.org/0000-0001-7691-4149
Valeria Mondelli http://orcid.org/0000-0001-8690-6839
Brandon A Kohrt http://orcid.org/0000-0002-3829-4820

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
