## [Reviewer comments · BMJ Open]

ARTICLE DETAILS

TITLE (PROVISIONAL)	Detection of risk for depression among adolescents in diverse global settings: Protocol for the IDEA qualitative study in Brazil, Nepal, Nigeria and the United Kingdom
AUTHORS	Wahid, Syed; Pedersen, Gloria; Ottman, Katherine; Burgess, Abigail; Gautam, Kamal; Martini, Thais; Viduani, Anna; Momodu, Olufisayo; Lam, Crystal; Fisher, Helen; Kieling, Christian; Adewuya, Abiodun O; Mondelli, Valeria; Kohrt, Brandon A.

VERSION 1 – REVIEW

REVIEWER	Cesar Escobar-Viera University of Pittsburgh, U.S.
REVIEW RETURNED	14-Apr-2020

GENERAL COMMENTS	General comments This protocol is entitled “Detection of risk for depression among adolescents in diverse global settings: Protocol for the IDEA qualitative study in Brazil, Nepal, Nigeria, and the United Kingdom.” While the protocol study is very clear and compelling, the goal of the protocolled study is much more complex than what the title suggests. Indeed, the study assesses cultural differences in conceptualization of depression, feasibility of a risk calculator, and feasibility of biological psychiatry research. While the manuscript at hand describes the focus on identification of risk factors and cultural differences in depression concepts, it does so without a clear theoretical/conceptual framework. Moreover, the protocol, loses focus on the risk detection tool (which goes largely unexplained), and biological psychiatry research. For these reasons, I cannot recommend publication of this protocol at this time. Major comments  1. Because of the methodology described, this study seems like it could be a big health services research study. However, it is hard to follow authors’ goals and objectives because the manuscript provides no theoretical/conceptual framework of the main constructs that guide this study. 2. Please clarify the role of surveys in the Delphi activities. As stated, this part of the methodology suggests a potential quantitative or mixed methods activity. Please clarify how this is not the case. 3. Setting descriptions are too long. If an appendix is allowed to describe each country’s description, I would recommend making this section shorter and expanding the descriptions in the appendix. 4. For KII and FGD, authors mention interview guides will be theory-informed. However, as I previously pointed out in (1), no theoretical/conceptual framework is provided. Therefore, it is unclear how theory is driving this part of the process. 5. Data collection and analysis for Delta activity is not described.
---

	6. It is not clear how country-specific data will be integrated in the analysis process.
--	--

REVIEWER	Ole Rikard Haavet University of Oslo, Norway
REVIEW RETURNED	16-Apr-2020

GENERAL COMMENTS	The protocol describes a study that can be of significant importance in an international perspective. The goal is to detect depression in young people at an early stage. Depression in adolescents seems to be a universal challenge. A strength of this study is the economic and cultural breadth In a report dated October 23, 2019, WHO stated the following: Half of all mental health conditions start at 14 years of age but most cases are undetected and untreated. The protocol is thoroughly prepared and there is a good connection between research questions and plans and methods for answering the questions. The research work is extensive with regard to data collection. The study uses Delphi method, theory of change and qualitative methods with key informants and focus groups. The study is conducted at a number of locations that cover variation in terms of culture and economic status. Both risk factors and protective factors will be investigated. The protocol describes a comprehensive but feasible study. I have a few comments. First, I think primary health care is lacking both in the research team and in the research chain. I think Mauerhoffer et al. (2009) describe very well that bodily ailments are often the gateway when depressed youth seek help. Furthermore, in an ongoing study, I have personally seen various instruments used to detect depression in adolescents. Questionnaire, to a small extent, helps us detect depressed adolescents, while the symptom anxiety seems to be able to increase the sensitivity to depression in this age group. Therefore, the symptom anxiety can be advantageously included in the study. With these two suggestions, I will support the publication of the protocol.
---

VERSION 1 – AUTHOR RESPONSE

Reviewer: 1

While the protocol study is very clear and compelling, the goal of the protocolled study is much more complex than what the title suggests. Indeed, the study assesses cultural differences in conceptualization of depression, feasibility of a risk calculator, and feasibility of biological psychiatry research. While the manuscript at hand describes the focus on identification of risk factors and cultural differences in depression concepts, it does so without a clear theoretical/conceptual framework. Moreover, the protocol, loses focus on the risk detection tool (which goes largely unexplained), and biological psychiatry research. For these reasons, I cannot recommend publication of this protocol at this time.

3. **AUTHORS' RESPONSE:** Thank you very much for these valuable comments. The conceptual framework and biological psychiatry components are addressed in Point#4 of our response below. For the risk detection tool, a description has been added as a Box in the key-informant interview and focus group discussion section.

“Specifically, we will gather stakeholder perspectives on the cultural acceptability and feasibility of the implementation of a risk calculator for adolescent depression (Please see Box-1).” (pg. 10)

Box 1: IDEA Study risk calculator

In regard to the study title, we thank the reviewer for this observation, and acknowledge the concern as valid. However, we also appreciate the need to balance for brevity, and therefore retain our original title which highlights the central topic of inquiry, namely depression risk in global settings – we hope this will be acceptable.

Major comments

1. Because of the methodology described, this study seems like it could be a big health services research study. However, it is hard to follow authors' goals and objectives because the manuscript provides no theoretical/conceptual framework of the main constructs that guide this study.

4. **AUTHORS' RESPONSE:** Thank you very much for this comment. We agree that the inclusion of a conceptual framework substantially clarifies how the study topics and methodologies are connected. We have included a conceptual framework and accompanying text to do so. The last sentence in the accompanying text (see below) clarifies the role of biological psychiatry research in the IDEA sites. Please see the additional file titled “Figure-1 Conceptual framework.jpg” for the conceptual figure. The following text accompanies the figure:

“The IDEA qualitative study is structured according to the social ecological model of health and Singer and Baer’s world system theory on the social origins of disease.[15, 16] Using these two guiding theoretical frameworks, we seek to understand the role of individual, interpersonal, institutional, community and policy factors and their interrelations, in depression risk and identification in adolescence (Please refer to Figure-1). Informed by George Engels’ classic model, we will elicit biopsychosocial risk and protective factors of depression within and between each ecological stratum.[17] We will utilize Kleinman’s Explanatory Model framework to explore the lived experience of depression at the individual level, including culturally driven local idioms of distress.[18, 19] We will further explore how these explanatory models are influenced by relationships at the interpersonal and primary group levels (family and friends), and cultural and social norms at the community level. At the institutional level, we will examine mental health services for depression identification and management, and acceptability and feasibility of risk detection at schools, primary health care, and social services. At the policy level, we will seek to understand challenges and opportunities to facilitate better depression detection and management. Additionally, in the LMIC sites, we will explore institutional capacity for conducting biological psychiatry research (i.e. biological specimen collection, storage, and testing capacity; and research capacity of universities and staff) and policy level considerations for ethical research governance that can support sensitive biological psychiatry research.” (pg. 7)

2. Please clarify the role of surveys in the Delphi activities. As stated, this part of the methodology suggests a potential quantitative or mixed methods activity. Please clarify how this is not the case.

5. **AUTHORS' RESPONSE:** The Delphi section has been revised to indicate that it will be a mixed-methods study. The role of qualitative interviews has been clarified.

“A Delphi panel consensus study using quantitative and qualitative methods will be conducted soliciting opinions from global experts in the field of adolescent depression.” (pg. 8)

“Following the 3 Delphi rounds, we will conduct in-depth interviews with panelists based on quantitative results. Panelists will provide comments on summary results tables explaining personal and panel rankings, interesting patterns in the quantitative results, and provide narratives on how cultural and contextual factors influence biopsychosocial risk factors. Qualitative data will be analyzed using thematic analysis.[32]” (pg. 9)

3. Setting descriptions are too long. If an appendix is allowed to describe each country's description, I would recommend making this section shorter and expanding the descriptions in the appendix.

6. AUTHORS' RESPONSE: Thank you for this suggestion. The country sections have now been moved to an appendix.

4. For KII and FGD, authors mention interview guides will be theory-informed. However, as I previously pointed out in (1), no theoretical/conceptual framework is provided. Therefore, it is unclear how theory is driving this part of the process.

7. AUTHORS' RESPONSE: We have reworded this to state that the guides were created based on the Conceptual Framework described in Point#4 of our response.

“We will first create an interview guide drawing from our conceptual framework and conduct approximately six preliminary KIIs with different stakeholders, across each site.” (pg. 12)

5. Data collection and analysis for Delta activity is not described.

8. AUTHORS' RESPONSE: The structure of the manuscript has been reorganized so that each method has clear sub-sections describing data collection and analysis processes. The quantitative data collection and analysis are clearly titled now. The text provided in Point#5 of our response describes the qualitative interviews.

6. It is not clear how country-specific data will be integrated in the analysis process.

9. AUTHORS' RESPONSE: We have added a few sentences at the end of the KII/FGD data analysis section that describes how we will manage country specific data.

“For each theme, we will present results from each site to facilitate comparison across countries. We will present similarities and differences in results, across stakeholders and across sites. If applicable, we will construct and present explanatory narratives for any heterogeneity in the results i.e. possibly attributing to cultural or contextual factors.” (pg. 14)

Reviewer: 2

The protocol describes a study that can be of significant importance in an international perspective. The goal is to detect depression in young people at an early stage. Depression in adolescents seems to be a universal challenge. A strength of this study is the economic and cultural breadth. In a report dated October 23, 2019, WHO stated the following: Half of all mental health conditions start at 14 years of age but most cases are undetected and untreated.

The protocol is thoroughly prepared and there is a good connection between research questions and plans and methods for answering the questions. The research work is extensive with regard to data collection. The study uses Delphi method, theory of change and qualitative methods with key informants and focus groups. The study is conducted at a number of locations that cover variation in terms of culture and economic status. Both risk factors and protective factors will be investigated.

The protocol describes a comprehensive but feasible study.

10. AUTHOR'S RESPONSE: We thank the reviewer for the positive evaluation of our protocol.

I have a few comments. First, I think primary health care is lacking both in the research team and in the research chain. I think Mauerhoffer et al. (2009) describe very well that bodily ailments are often the gateway when depressed youth seek help.

11. **AUTHORS' RESPONSE:** Thank you very much for this comment. We have added information explicitly highlighting the importance of primary health care physicians and included the very helpful Mauerhoffer article as a citation.

“The inclusion of mental health specialists in the IDEA study is foundational, as these stakeholders are pivotal in providing care to depressed adolescents. However, as adolescents often present to primary care with depression, specifically with somatic symptomology, the inclusion of pediatricians and primary healthcare physicians are essential, especially in settings with diminished numbers of, and limited access to, mental health specialists.[40]” (pg. 12)

Furthermore, in an ongoing study, I have personally seen various instruments used to detect depression in adolescents. Questionnaire, to a small extent, helps us detect depressed adolescents, while the symptom anxiety seems to be able to increase the sensitivity to depression in this age group. Therefore, the symptom anxiety can be advantageously included in the study.

12. **AUTHORS' RESPONSE:** We have added information on how anxiety and depression often present as comorbid (and included a citation) and that we are embracing an open ended approach to exploring the signs and symptoms of depression during data collection, as narrated by the respondent.

“The experience of depression is heterogenous across the world.[37] Symptoms of anxiety disorder often co-present or can play a role in the onset of depression.[38] Accordingly, in the interviews, we will explore a range of negative affective symptoms by adopting an open-ended approach to probing the signs and symptoms of adolescents' experience of depression.” (pg. 10)

With these two suggestions, I will support the publication of the protocol.

13. **AUTHOR'S RESPONSE:** We thank the reviewer for the helpful suggestions and hope we have addressed them satisfactorily.

VERSION 2 – REVIEW

REVIEWER	Cesar Escobar-Viera University of Pittsburgh United States
REVIEW RETURNED	28-May-2020
GENERAL COMMENTS	I thank the authors for addressing my comments. I am excited about seeing this protocol published and even more curious about the reading the results of this important research.